# Changes Overtime in Perinatal Management and Outcomes of Extremely Preterm Infants in One Tertiary Care Romanian Center

**DOI:** 10.3390/medicina58081019

**Published:** 2022-07-29

**Authors:** Diana Ungureanu, Nansi S. Boghossian, Laura Mihaela Suciu

**Affiliations:** 1Neonatology Department, Maternity Hospital “Ioan Aurel Sbârcea”, 500025 Brasov, Romania; dianaungureanu75@yahoo.com; 2Department of Epidemiology and Biostatistics, Arnold School of Public Health, University of South Carolina, Columbia, SC 29208, USA; nboghoss@email.sc.edu; 3Department of Neonatology, University of Medicine Pharmacy Science and Technology “George Emil Palade” of Târgu Mureș, 540139 Târgu Mureș, Romania

**Keywords:** antenatal management, extremely preterm newborns, neonatal morbidities, neonatal mortality

## Abstract

*Background and Objectives***:** Extremely preterm infants were at increased risk of mortality and morbidity. The purpose of this study was to: (1) examine changes over time in perinatal management, mortality, and major neonatal morbidities among infants born at 25^0^–28^6^ weeks’ gestational age and cared for at one Romanian tertiary care unit and (2) compare the differences with available international data. *Material and Methods*: This study consisted of infants born at 25^0^–28^6^ weeks in one tertiary neonatal academic center in Romania during two 4-year periods (2007–2010 and 2015–2018). Major morbidities were defined as any of the following: severe intraventricular hemorrhage (IVH), severe retinopathy of prematurity (ROP), necrotizing enterocolitis (NEC), and bronchopulmonary dysplasia (BPD). Adjusted logistic regression models examined the association between the mortality and morbidity outcome and the study period. *Results*: The two cohorts differed with respect to antenatal antibiotics and rates of cesarean birth but had similar exposure to antenatal steroids and newborn referral to the tertiary care center. In logistic regression analyses, infants in the newer compared to the older cohort had a lower incidence of death (OR: 0.19; 95% CI: 0.11–0.35), a lower incidence of IVH (OR: 0.26; 95% CI: 0.15–0.46), and increased incidence of NEC (OR: 19.37; 95% CI: 2.41–155.11). *Conclusions:* Changes over time included higher use of antenatal antibiotics and cesarean delivery and no change in antenatal steroids administration. Overall mortality was lower in the newer cohort, especially for infants 25^0^–26^6^ weeks’ gestation, NEC was higher while BPD and ROP were not different.

## 1. Introduction

Infants born very preterm, before 28 weeks of gestation, face the highest risks of neonatal mortality and morbidities, as well as long-term neurodevelopmental impairment [1,2]. Survival of very preterm infants has increased over time, and within each week of gestational age, due to advances in prenatal, perinatal, and neonatal care [3]. Recent studies have shown wide variation in outcomes of very preterm infants among countries with comparable levels of development and health care systems, raising questions about the population and health system factors that may influence preterm birth outcomes [4]. Survival of very preterm infants born before 28 weeks of gestation is over 80% in North America [5,6], Australia [7], Japan [8], and Switzerland [9]. Neonatal mortality in developing European countries is still somewhat higher than in Western Europe. In a Lithuanian cohort of preterm infants born at ≤27 weeks, neonatal mortality was 50% [10], and in an Estonian report, neonatal mortality at ≤25 weeks decreased from 59% in 2002–2003 to 46% in 2007–2008 [11]. In Romania, survival to discharge among preterm infants born at ≤28 weeks has improved between 2007–2010 and 2011–2014 from 34% to 70% [12]. Decision-making at these gestations is an important determinant of both mortality and morbidity, with substantial international variability in the antenatal and postnatal management of threatened extremely preterm deliveries [13].

In this study, we examined changes over time (2007–2010 vs. 2015–2018) in perinatal management, survival, and major morbidities among 25^0^–28^6^-week gestation extremely preterm babies born at or cared for at one Romanian tertiary care unit.

## 2. Materials and Methods

We conducted this prospective study on infants born between 25^0^- and 28^6^-weeks’ gestation at one of the ten tertiary Neonatal Intensive Care Units (NICUs) located in Romania comparing the periods between 1 January 2007–31 December 2010 (hereafter labeled older cohort) and 1 January 2015–31 December 2018 (labeled newer cohort). The University Hospital Brasov committee for human study research approved the use of maternal and neonatal data of admitted cases for this analysis. Women were informed that their records could be used for the evaluation of the medical practices and told they could choose to opt out of these studies. All approached women gave their written consent to participate.

In this tertiary center, the total number of admitted infants born before 37 completed weeks of gestation was 2492 of 19,173 newborns (12.9%) in the older cohort, and 2245 of 16,891 newborns (13.2%) in the newer cohort.

Among 25^0^–28^6^-week infants, we excluded home or ambulance deliveries (*n* = 12), babies with genetic anomalies (*n* = 16), babies not surviving to be admitted to the NICU (*n* = 23), and outborn infants admitted after 12 h of birth (*n* = 18), resulting in a total of 259 infants (120 infants in the 2007–2010 cohort and 139 infants in the 2015–2018 cohort). Among 23 newborns not surviving to be admitted to the NICU, 18 (78%) were born between 2007 and 2010, and 5 (21%) were born between 2015 and 2018.

All relevant ante-, peri-, and postnatal data were retrieved from an electronic database and included information about the course of pregnancy as well as the delivery, perinatal maternal infectious parameters (amniotic infection), premature rupture of membranes (PROM), peri- and neonatal information, the clinical course and major morbidities, and parameters related to respiratory support (e.g., continuous positive airway pressure, mechanical ventilation). Markers of a proactive approach to perinatal and neonatal management included obstetric interventions (antenatal corticosteroids and cesarean delivery) and neonatal interventions (surfactant, respiratory support, neonatal intensive care unit admission) [14].

Study variables. To describe and compare the population during the two cohorts, we collected the following data, which were the same for the two time periods: gestational age (GA) was determined by the best estimate based on first-trimester ultrasound, the last menstrual period, or neonatal examination after birth, in that order. Antenatal transfer was defined as transfer from primary or secondary level care center between booking and delivery. Transfers from Level I or Level II were combined into the same group given the small number of mothers transferred. PROM was defined as cervicovaginal discharge that was confirmed by sterile speculum examination and low index of amniotic fluid on ultrasound examination; the latency period was the time interval between PROM and birth [15]. Chorioamnionitis was diagnosed by the presence of maternal fever (body temperature > 37.8 °C) accompanied by two or more of the following criteria: (1) uterine tenderness, (2) foul-smelling amniotic fluid, (3) fetal tachycardia (fetal heart rate > 160 beats/min), (4) maternal tachycardia (heart rate > 100 beats/min), and (5) maternal leukocytosis (leucocyte count > 15,000 cells/mm^3^) [16]. Antenatal corticosteroids, antenatal antibiotics, cesarean delivery, surfactant therapy, and respiratory support were reported because they are associated with neonatal outcomes. Surfactant was administered with the patient intubated. Novel noninvasive surfactant administration techniques were not available at the time of the study period in this center.

Mothers were considered to have received antenatal corticosteroids if they received a full course of 4 doses of dexamethasone (betamethasone was not used). Antenatal antibiotic treatment was recorded if any antibiotic drug was administered to the mother during the same admission in which the birth took place. Small for gestational age (SGA) was defined as sex-specific birth weight below the 10th percentile for the gestational age based on the Fenton Preterm Growth Chart [17]. The delivery mode recorded was vaginal or cesarean. Multiple births included twins, as no higher-order multiples were recorded during the study period.

Outcome measures. The primary outcome measure was neonatal mortality, and the secondary outcome was major morbidities. We analyzed survival in 1-week gestational age increments. Severe neonatal morbidities included: severe bronchopulmonary dysplasia (BPD), severe intraventricular hemorrhage (IVH), necrotizing enterocolitis (NEC), and severe retinopathy of prematurity (ROP). BPD was defined by the need for oxygen therapy and/or mechanical ventilatory support (endotracheal or noninvasive) at 36 weeks [18]. IVH included stages ≥ III, associated with a hemorrhagic lesion of the adjacent parenchyma [19]; cerebral lesions were mostly diagnosed using a cranial ultrasound, which is standard care in Romania. The first screening ultrasound was performed ordinarily on or approximately the seventh day of life, and the second ultrasound was performed for all infants before discharge. For those infants who died before the seventh day of life, hemorrhage was identified on emergency ultrasound whenever indicated or by necropsy. ROP included stages ≥ III or any stage associated with plus disease requiring cryotherapy or laser photocoagulation of the peripheral avascular retina https://www-sciencedirect-com.am.e-nformation.ro/topics/medicine-and-dentistry/retinopathy-of-prematurity (accessed on 19 October 2020) according to the international classification [20] and Early Treatment for ROP (ETROP) trial [21].

Statistical analyses. To determine statistical significance for unadjusted comparisons we used χ^2^ or Fisher’s exact test. Maternal and neonatal characteristics between the two epochs were compared using the Mantel–Haenszel chi-square test for categorical variables and the Kruskal–Wallis test for continuous variables. Continuous variables are reported as medians ± standard deviation (SD). Logistic regression tests included Hosmer and Lemeshow, -2Log likelihood, Cox & Snell R Square, and Nagelkerke R Square tests. All tests were 2-tailed; to detect statistical significance *p* < 0.05 was used. Multivariate logistic regression models were conducted to examine the associations between the time period and the outcomes of mortality and major morbidities, adjusting for gestational age, SGA, twins, and male sex. All statistical analyses were conducted by using SAS version 9.3 (SAS Institute Inc., Cary, NC, USA).

## 3. Results

### 3.1. Study Population and Perinatal Interventions

A total of 120 infants in the older cohort and 139 infants in the newer cohort were eligible for analysis. Univariate comparison revealed that mothers in the older compared to the newer cohort were more likely to have chorioamnionitis and less likely to receive antenatal antibiotics and deliver via cesarean (Table 1). Antenatal steroids and tocolysis were not more frequent in the newer cohort. Fetal presentation vertex and multiple pregnancies were similarly distributed between the two epochs. Additionally, compared to 12 (9%) maternal transfers in the older cohort, there were none in the newer cohort (*p* = 0.001). Postnatal transfers were higher in the newer compared to the older cohort (14.3% vs. 9%); however, the difference was not statistically significant (*p* = 0.19). Interestingly, a higher rate of antenatal antibiotics was found in the newer compared to the older cohort (45% vs. 28%) despite a lower rate of chorioamnionitis in the newer cohort (4% vs. 19%), and these differences were statistically significant (*p* < 0.05).

### 3.2. Neonatal Characteristics

There were no differences between the two epochs in the number of infants born SGA and with a low Apgar score at 5 min, but there was a higher proportion of infants with low blood cord pH in the older cohort. In the newer cohort, a significantly higher proportion of infants were intubated and received surfactant (40% vs. 10%, *p* = 0.01), spent more days (median ± IQR) on respiratory support (3 ± 5 vs. 10 ± 18 days, *p* = 0.01), and were admitted longer to the NICU (4 ± 13 vs. 27 ± 35 days, *p* = 0.01). The number of hospital days (median ± IQR) among infants in the newer cohort was significantly higher (71 ± 72 vs. 40 ± 62 days, *p* = 0.01) compared to the older cohort.

### 3.3. Neonatal Mortality

The trends in mortality and antenatal interventions, including cesarean delivery, antenatal steroids, and antibiotics among different gestational ages, are displayed in Figure 1. In this graph, there is a trend towards increased cesarean delivery rates at 26-, 27-, and 28-weeks’ gestation in the newer than in the older cohort, which is also reflected in a decreased trend in mortality rates.

As expected, mortality rate decreased with increasing gestational age. However, survival was higher at all gestational ages in the newer than in the older cohort. The survival for those born at 25 weeks was 60% in the newer cohort compared to 20% in the older cohort. The overall mortality in the first 72 h of age decreased from 50% to 17% (*p* < 0.001) in the newer cohort. Mortality rates according to gestational age, birth year, and epoch are represented in Figure 2a,b.

### 3.4. Neonatal Morbidities

Unadjusted comparison of outcomes revealed significantly lower rates of mortality and IVH and higher rates of NEC, but not significantly different rates of ROP and BPD in the newer compared to the older cohort (Table 2).

We ran a binary logistic regression model; NEC was the dependent variable, and the antenatal antibiotics, year of birth, latency interval, and gestational age were included as covariates. Good model fit was evident by a nonsignificant Hosmer–Lemeshow test (*p* = 0.87). The analysis revealed higher values for each 1-year increase in study period (OR 1.14 (95% CI:1.14–1.84)), but not significant differences for the antenatal antibiotics (OR 1.51 (95% CI:0.5–4.5)), latency interval (OR 0.99 (95% CI:0.99–1.01)), or for each 1-week increase of the gestational age (OR 0.86 (95% CI 0.54–1.36)). Logistic regression analysis controlling for sex, SGA, twins, gestational age, and cohort epoch revealed significantly lower odds of mortality (OR 0.19 (95% CI: 0.11–0.35)) and severe IVH (OR 0.26 (95% CI: 0.15–0.46)) (Table 3).

## 4. Discussion

This study of outcomes of extremely preterm infants admitted to one neonatal tertiary intensive care Romanian unit is, to the best of our knowledge, the first comprehensive review to evaluate how care practices, major morbidity, and mortality have evolved over more than a decade.

We observed in our study important changes in the antenatal management of pregnant women at risk of preterm birth. The number of in utero transfers decreased (from 9% to 0%) without a marked change in postnatal transfers (9% vs. 14%). This finding is consistent with the changes in European perinatal activity centers after implementing guidelines for the care of infants born at the limit of viability [22,23]. The higher use of cesarean delivery and antenatal antibiotics indicate a more proactive approach to managing these pregnancies and could be related to the lower rates of mortality in the later cohort. A U.S. study showed that active treatment, defined as having received surfactant therapy, tracheal intubation, ventilatory support, parenteral nutrition, epinephrine, or chest compressions, accounted for 78% of the between-hospital variation in survival of infants born at 22 or 23 weeks of gestation [24]. A European study of neonates born below 500 g, has also shown that after PROM, neonatal survival was more favorable for mothers who received antenatal antibiotics and who delivered via a cesarean section [25].

Antenatal steroid use did not differ between the two periods. Corticosteroid administration rates were higher than 60% only at gestational ages above 26 weeks and just in the later cohort. Recommendations regarding administration of antenatal corticosteroids for the acceleration of fetal lung maturity to women at risk for preterm delivery in the early 1990s led to a reduction in mortality of preterm infants [26]. Our rate of antenatal steroids (overall around 30%) is surprisingly low compared to the US [27,28] rate, which has been reported to be more than 80% among 25-week infants [28]. Additionally, among US centers participating in the NICHD Neonatal Research Network, the administration rate of steroids increased each year, especially among mothers of low-gestational-age pregnancies [27], a trend we did not observe in our analyses. The design of our study did not allow us to examine the reasons for the low rate of antenatal corticosteroid administration. Common reasons include (i) the mother presenting late in labor [29], or (ii) an emergency cesarean section was performed due to fetal distress. Future studies should document these reasons and aim to improve the rate of antenatal corticosteroid administration, especially given the more proactive approach we observed in managing these high-risk pregnancies and the documented significant improvement in survival with a concordant receipt of antenatal steroids and postnatal life support [30].

Similar to the increase in antenatal management, postnatal management also increased in the newer cohort; surfactants were administered more frequently, and respiratory support and length of stay in both the neonatal intensive care unit and hospital also increased.

We noted a significant increase in the number of severe morbidities during the second period. For severe IVH, the risk was higher for those at the lowest gestational age, and this relation has also been observed in a previous study [31]. The incidence of BPD in the later cohort could be explained by the higher number of days on ventilatory support and the higher frequency of surfactant administration, although they both may be associated with increased survival in this vulnerable population [32,33]. It must be noted that we did not demonstrate increasing risk of BPD with decreasing GA; however, our relatively small number of surviving infants at each gestational age week in this single-center study may have prevented finding an association of decreasing BPD risk with increasing gestational age [34].

The newer cohort had a significantly increased incidence of NEC compared to the older cohort. Most infants were fed with their mother’s own milk, and this practice did not change during the studied period. This center keeps the mothers hospitalized while their newborn is admitted to the NICU specifically for feeding purposes. However, in a few cases, the newborn was fed by formula adapted for preterm infants when the mother’s milk was not available. The dramatically increased incidence of NEC noted in the newer cohort may be partly explained by increased intensive care and increased survival as published in a U.S. study [27]. Given the fact that our cohort has a small sample size, we cannot make a conclusion about the relationship between increased survival and avoidable severe morbidities, i.e., NEC, or provide definitive reasons for the observed change.

Our study is limited to one Romanian center, potentially limiting the generalizability of our findings. However, we provided much-needed information on the perinatal management and outcomes of these infants in Romania. Our study was also limited by the inability to evaluate the placental pathology that could have provided important information to ascertain the cause of preterm births and adverse neonatal outcomes, especially at gestational ages lower than 28 weeks [35].

## 5. Conclusions

We found increased rates of cesarean section and antenatal antibiotics and duration of respiratory support, implying a more proactive approach to managing infants in the newer cohort. However, we cannot imply that the association between obstetric and neonatal interventions and mortality and morbidity in these lower gestational age preterm infants is causal. In the current study, several factors did not improve over time. Administration of corticosteroids remains low. It is possible that many mothers present late in labor, and there may be limited opportunity for obstetricians to administer steroids. The increased survival rates were accompanied by longer hospitalization, longer duration of respiratory support, and higher rates of NEC but not ROP and BPD.

## Figures and Tables

**Figure 1 medicina-58-01019-f001:**
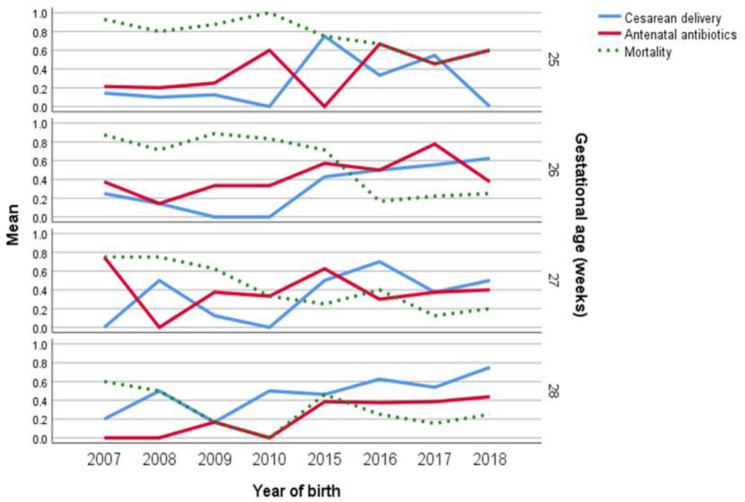
Relation between antenatal interventions and overall mortality among different gestational ages according to the year of birth.

**Figure 2 medicina-58-01019-f002:**
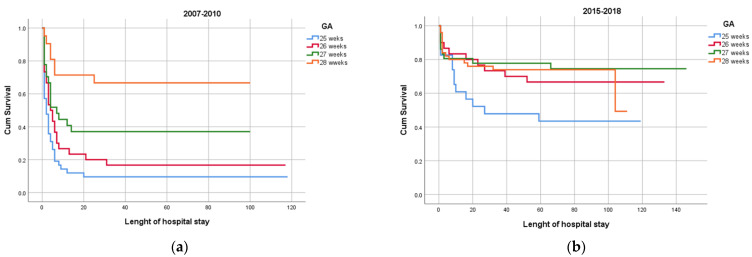
(**a**) Survival to discharge according to gestational age in older cohort and (**b**) newer cohort.

**Table 1 medicina-58-01019-t001:** Baseline characteristics of the mothers and their newborns according to study period.

Variables	Older Cohort N = 120 *n* (%)	Newer Cohort N = 139 *n* (%)	Overall N = 259 *n* (%)	*p* Value ^b^
Maternal characteristics				
Maternal age mean (SD), years	27.9 (6.7)	29.1 (7.1)	28.59 (7.0)	0.1
Maternal referral to tertiary care	12 (9.0)	0	12 (5.0)	0.001
Maternal hypertensive disorder ^a^	7 (6.0)	10 (7.0)	17 (7.0)	0.6
Gestational diabetes ^a^	0	1 (1.0)	1 (0.1)	0.3
Chorioamnionitis	23 (19.0)	5 (4.0)	28 (11.0)	0.001
Tocolysis	22 (18.0)	25 (18.0)	47 (30.0)	1.0
Antenatal steroids	35 (29.0)	43 (31.0)	78 (30.0)	0.7
Antenatal antibiotics	33 (28.0)	62 (45.0)	95 (37.0)	0.004
PROM	46 (38.0)	50 (36.0)	96 (37.0)	0.2
PROM latency interval, days Median (IQR)	4 (12)	1 (12)	1 (12)	0.6
Cesarean section	18 (15.0)	75 (54.0)	93 (36.0)	0.001
Newborn characteristics				
Male	59 (49.0)	76 (55.0)	135 (52.0)	0.3
Fetal presentation vertex	78 (65.0)	81 (58.0)	139 (53.6)	0.2
Twins	22 (18.0)	31 (22.0)	53 (20.5)	0.4
Apgar score at 5 min < 7	92 (77.0)	92 (66.0)	184 (71.0)	0.06
Small for GA, <10th percentile on Fenton	25 (21.0)	29 (21.0)	54 (21.0)	0.9
Blood cord pH < 7	30 (25.0)	12 (8.6)	42 (16.2)	0.001
Gestational age				0.001 ^b^
25 weeks	42 (35)	23 (16.5)	65 (25.1)	
26 weeks	30 (25)	30 (21.6)	60 (23.2)	
27 weeks	27 (22.5)	36 (25.9)	63 (24.3)	
28 weeks	21 (17.5)	50 (35.9)	71 (27.4)	
Gestational age at admission, median (IQR)	26.2 (1.1)	26.8 (1.1)	26.0 (1.5)	0.001
Birth weight mean (SD), grams	759.7 (162)	951 (231)	862 (223)	0.001
Newborn referral to tertiary care	11 (9.0)	20 (14.3)	31 (12.0)	0.1
Surfactant administration	26 (21.6)	104 (75.0)	130 (50.0)	0.001
Length of respiratory support, days, median (IQR)	3 (5)	10 (18)	5 (10)	0.001
Length of NICU stay, days, median (IQR)	4 (13)	27 (35)	14 (31)	0.001
Length of hospital stay, days, median (IQR)	40 (62)	71 (72)	38 (81)	0.001

^a^ 16 cases (nine cases in older cohort and seven cases in newer cohort) were missing information about history of chronic hypertension, hypertensive disorders of pregnancy, and gestational diabetes; these cases were included in the analysis. ^b^ Chi-square test or Fisher’s exact test, as appropriate: global test comparing distributions between older and newer cohorts. Numbers are N (%) unless otherwise stated.

**Table 2 medicina-58-01019-t002:** Comparison of perinatal outcomes between two study periods.

Variables	Older Cohort N = 120 *n* (%)	Newer Cohort N = 139 *n* (%)	Overall N = 259 *n* (%)	*p* Value ^a^
IVH grade > III	73 (60.9)	36 (25.9)	109 (42.0)	0.001
Mortality in the first 72 h age	60 (50.0)	24 (17.2)	84 (32.4)	0.001
Mortality overall				
25 weeks	38/42 (90.5)	13/23 (56.5)	51/65 (78.0)	0.001
26 weeks	25/30 (83.3)	10/30 (33.3)	35/60 (58.3)	0.28
27 weeks	17/27 (62.9)	9/36 (25.0)	26/63 (41.2)	0.08
28 weeks	7/21 (33.3)	14/50 (28.0)	21/71 (29.5)	0.001
Infants who survived to discharged	33 (27.5)	93 (66.9)	126 (48.7)	0.001
^b^ BPD	11 (33.3)	31 (33.3)	42 (33.3)	0.9
^b^ ROP > III	10 (33.3)	26 (27.9)	36 (28.6)	0.4
^b^ NEC	1 (3.03)	16 (17.2)	17 (13.5)	0.001

^a^ Chi-square test or Fisher’s exact test, as appropriate: global test comparing distributions between older and newer cohorts. ^b^ Reported numbers and percentages for survivors to hospital discharge. Numbers are N (%) unless otherwise stated.

**Table 3 medicina-58-01019-t003:** Odds of mortality and IVH grade III or higher by study period and gestational age.

	Mortality Older Cohort OR 95% CI	Mortality Newer Cohort OR 95% CI	IVH Grade > III Older Cohort OR 95% CI	IVH Grade > III Newer Cohort OR 95% CI
25 weeks	5.62 (1.82–17.37)	3.27 (1.31–8.18)	2.88 (1.25–6.67)	4.18 (1.64–10.63)
26 weeks	2.25 (0.78–6.51)	1.01 (0.43–2.39)	1.39 (0.58–3.32)	0.84 (0.32–2.16)
27 weeks	0.55 (0.22–1.39)	0.59 (0.25–1.39)	0.75 (0.31–1.79)	0.48 (0.18–1.28)
28 weeks ^a^	Reference	Reference	Reference	Reference

Models adjusted for sex, twins, SGA, gestational age, and cohort epoch. ^a^ 28 weeks is the reference group.

## Data Availability

The data that support the findings of this study are available from the corresponding author L.M.S. upon reasonable request.

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
