# Peer review of "Changes Overtime in Perinatal Management and Outcomes of Extremely Preterm Infants in One Tertiary Care Romanian Center"

_medicina, 2022, doi:10.3390/medicina58081019_

Round 1

Reviewer 1 Report

The authors present here a well designed study to evaluate the changes over time the outcomes of extremely preterm newborns admitted to a tertiary level NICU in Romania. The article is well written, however, minor changes are required before the article is accepted for publication.

-Abstract:

Line 22: outcome and the study periods.

Line 25-26: Recommend replacing 'risk' with 'incidence' at all 3 places

-Introduction:

The authors could perhaps comment on the outcomes of preterm births <29 weeks in developing countries as that would be more comparable to the current study.

- Materials and Methods:

Lines 76-78: Markers of proactive approach...... include obstetricians obstetric interventions (antenatal....) and neonatologist neonatal interventions...

Line 95: Surfactant has been  was..

-Results:

Lines 139-141: Suggest reframing as follows:

"Additionally, compared to 12 (9%) maternal transfers in the older cohort, there were none in the newer cohort (p=0.001). Postnatal transfers were higher in the newer cohort compared (14.3% vs 9%), however, the difference was not statistically significant (p=0.19)."

Line 149. Would suggest that that authors mention an interestingly higher rate of antenatal antibiotic use the newer cohort despite a lower rate of chorioamnionitis.

Line 166: in the newest newer than in the older cohort.

Table 2: I recommend including p values for mortality for different gestational ages in addition to that for overall mortality.

Table 3 is very confusing. I recommend that that authors redo the table. Did the authors compare mortality and IVH>grade III for gestational age 25, 26 and 27 weeks to that of infants born at 28 weeks? If yes, It may be prudent to calculate the OR for the 2 cohorts separately.

for example, comparing mortality in 25 weeker infants to 28 weeker infants in the older and newer cohorts separately.

Since the rate of chorioamnionitis was higher in the older cohort with a lower incidence of antenatal Antibiotics use, It would be of interest to the readers to see if there was a difference in sepsis rates amongst the two cohorts.

-Discussion

 The authors must be more clear in the discussion that the higher length of respiratory support and hospital stay was likely associated with a more proactive obstetric as well as neonatal management of these infants.

Similarly, in conclusion, this should be emphasized as well.

The newer cohort had a significantly increased incidence of NEC compared to the older cohort. Was there a change in feeding guidelines? Can the authors attempt to explain why such a dramatic increase in NEC was noted in the newer cohort?

Reviewer 2 Report

The study itself is excellently designed and done. Properly methodologically done observation of the same parameters through two times

Unfortunately, nothing new has happened in Neonatology in the last 20 years, except the use of Surfactant and better management of extremely immature children on mechanical ventilation / thanks to advances in technology

Author Response

On behalf of the authors of the ID Medicina 1786186 I thank you for a very thoughtful review of the manuscript. The manuscript was checked again for English language. We hope you will find this revised manuscript suitable for publication, the paper is much improved and clearer thanks to the suggestions of the reviewer and editor.